# Characterizing Repellencies of Methyl Benzoate and Its Analogs against the Common Bed Bug, *Cimex lectularius*

**DOI:** 10.3390/insects13111060

**Published:** 2022-11-17

**Authors:** Jaime Strickland, Nicholas R. Larson, Mark Feldlaufer, Aijun Zhang

**Affiliations:** Invasive Insect Biocontrol and Behavior Laboratory, Beltsville Agricultural Research Center-West, USDA-ARS, Beltsville, MD 20705, USA

**Keywords:** natural product, methyl benzoate, methyl benzoate analogs, DEET, repellent, EthoVision

## Abstract

**Simple Summary:**

The resistance of bed bugs to many of the marketed insecticides has contributed to the recent resurgence in bed bug infestations. This study tests methyl benzoate and several of its analogs for repellency against the bed bug species, *Cimex lectularius.* It was found that many benzoate compounds exhibit repellency against bed bugs, with naturally occurring volatile aroma compounds methyl 2-methoxybenzoate (M2MOB) and methyl 3-methoxybenzoate (M3MOB), exhibiting the longest-lasting repellency against both insecticide susceptible and pyrethroid resistant strains of bed bug.

**Abstract:**

Bed bug infestations are on the rise globally, and remediation efforts are becoming more expensive and difficult to achieve due to rising insecticide resistance in the pest populations. This study evaluates *Cimex lectularius* behavior in the presence of attractive elements—aggregation pheromone or food source (human blood)—and the reported botanical repellent methyl benzoate (MB), several MB analogs, as well as the well-known insect repellent, *N*,*N*-diethyl-meta-toluamide (DEET). Utilizing EthoVision, a video tracking system, we now report that MB and several of its analogs exhibit strong spatial repellency against *C. lectularius,* with methyl 2-methoxybenzoate (M2MOB) and methyl 3-methoxybenzoate (M3MOB) exhibiting the strongest repellent effects. Further, our data showed that MB, M2MOB, M3MOB, and DEET exhibit repellency against a pyrethroid resistant strain of *C. lectularius.*

## 1. Introduction

The common bed bug, *Cimex lectularius* L., has been a nuisance to humans for thousands of years [1]. Bed bugs prefer to feed on human blood and can cause health problems, including psychological distress [2], allergic symptoms, and skin rashes [3]. Although the number of bed bug cases in North America and Western Europe dropped by the end of the 20th century, they have since begun to reemerge [4,5]. Despite a long history of experience, bed bug infestations remain difficult to treat. To remove an existing bed bug infestation, integrated pest management programs involving chemical (desiccant dusts, insecticides) and non-chemical (heating, freezing, removing mechanically) treatments are often employed [6,7,8,9,10], but these can be expensive, difficult, time consuming, and potentially damaging to belongings [11]. A study comparing two different integrated pest management strategies (desiccant dust and insecticide) in low-income housing in 2005 and 2007 cites an average cost of $463–482 (USD) per apartment for treatment [12], while whole house treatments can be $2000–4000 or more depending on location and infestation level [13]. Though an exact dollar amount is difficult to ascertain, when costs associated with hospitality and travel [14,15,16], legal expenses [17,18,19,20], retail expenses, and brand damage are all considered, the sum total of damages caused by bed bugs can be estimated to be on the order of billions of dollars (USD) annually [21]. Due to the difficulty and cost associated with treating infestations, the ability to prevent infestations is invaluable [22].

The resurgence of bed bug infestations in developed countries during the late twentieth century is often attributed to the increase in world travel [4,5,23], so preventative measures, such as protecting personal belongings with protective linings [24] or repellents [22,25,26,27,28,29,30], are an important area of research. Several commercial products are marketed as being bed bug repellents (e.g., Bed Bug Killer, Ready-To-Use Spray by Nature’s Mace) and treating belongings prior to travel is recommended as a technique to reduce risk of acquiring bed bugs. Compounds including *N*,*N*-diethyl-meta-toluamide (DEET), *N*,*N*-diethyl phenyl acetamide (DEPA), isolongifolenone (USDA patented insect repellent), and several others have been shown to exhibit repellency in mosquitoes, ticks, and bed bugs [25,31]. Pyrethroid insecticides (such as deltamethrin and lambda-cyhalothrin) have been the primary chemical control of bed bugs globally for decades, however with the development of widespread resistance their efficacy is waning [32]. The difficulty associated with preventing bed bug transportation highlights the value of discovering new chemical repellents. 

Previous studies have shown that a botanical compound, methyl benzoate, exhibits toxicity and repellency [33,34] against several insect pests, including *Drosophila suzukii*, *Halyomorpha halys*, *Plutella xylostella*, *Manduca sexta* [34], *Spodoptera frugiperda* [35], *Solenopsis invicta* [36], some stored product insect pests [37], some post-harvest insect pests [38], and the bed bug, *Cimex lectularius* [39]. The present study builds off this work by further investigating the repellent properties of methyl benzoate and several of its derivatives against bed bugs. Various aspects of the compounds were explored including (1) longevity of spatial repellency of individual compound, and (2) whether spatial repellency was maintained within the presence of attractants in pesticide susceptible and resistant strains.

## 2. Materials and Methods

### 2.1. Insects

A colony of insecticide susceptible *C. lectularius* was established from bugs originally obtained from Harold Harlan (Crownsville, MD, USA) [40], and a colony of pyrethroid resistant *C. lectularius* was established from bugs originally obtained from the University of Massachusetts. For the resistant population, resistance was verified by applying permethrin at increasing doses, up to 500 times the susceptible strain LD50 to the bed bugs’ ventral side. Three sets of ten insects were tested with zero deaths observed at the maximum tested dose. These colonies were maintained at ambient conditions (25 ± 2 °C and 40 ± 15% relative humidity [RH]) and fed weekly on expired human red blood cells and plasma (1.25:1 *v*/*v*), received from the Walter Reed National Military Medical Center, Bethesda MD, using an artificial (in vitro) feeding system as previously described [41]. For all experiments, an equal number of males and females were used for each condition tested.

### 2.2. Chemicals

Methyl benzoate (MB), Methyl 2-chlorobenzoate (M2CB), Methyl 2-methylbenzoate (M2MB), Methyl 2-methoxybenzoate (M2MOB), Ethyl benzoate (EB), *N*,*N*-diethyl-meta-toluamide (DEET), Acetone (solvent), (*E*)-2-Hexenal, and (*E*)-2-Octenal were purchased from Sigma Aldrich (St. Louis, MO, USA). Methyl 3-methylbenzoate (M3MB), Methyl 3-methoxybenzoate (M3MOB), and Vinyl benzoate (VB) were purchased from Fisher Scientific (Waltham, MA, USA). Hexyl benzoate (HB) was purchased from Alfa Aesar (Ward Hill, MA, USA). Purity of all chemicals was ≥95%. Chemical structures and CAS registry numbers for all compounds are shown in Figure 1. 

### 2.3. Experimental Setup

#### 2.3.1. Repellency Longevity

The experimental setup (Figure 2) was modified from previous studies that utilized video tracking software to test spatial repellency in bed bugs [39]. Assays were performed in a darkened room under a Basler ACE acA 1300–60 gm high-resolution monochrome camera (Basler AG, Ahrensburg, Germany). EthoVision^®^ XT10 (Noldus Information Technology, Wageningen, the Netherlands) video tracking software was utilized to capture the behavioral movements of the individuals. The arena was illuminated by infrared light emitted by an Axton AT-8 infrared LED illuminator (AxtonTech, North Salt Lake, UT, USA).

Ten microliters of pure test compound were applied to a 6 mm filter paper disc (Whatman, Grade AA) and allowed to evaporate for a designated amount of time of 0 h, 24 h, and 7 days for all compounds. For compounds that showed repellency at 7 days, further tests were performed weekly until repellency was lost, up to 28 days. After the evaporation period, the test disc was then placed into a 150 mm glass Petri dish top arena that was lined with filter paper (Whatman #1 qualitative 150 mm), with the center of the disc approximately one centimeter from the wall in the top position. Three untreated filter paper discs were then placed evenly around the perimeter of the arena in the right, bottom, and left positions. This setup has been previously shown to not have a positional bias [39], and to produce heatmap figures, the treated test disc was placed in the same position for each replicate performed. Further, a blank control (no treatments on any disc) was included to confirm that there was not a positional bias. A clean Petri dish top lined with new filter paper was used for each replicate. The absorbent discs were replaced for each trial as well. An individual *C. lectularius* was placed in the center of the arena, equidistant from the four filter paper discs and covered with a pipet tip for five minutes to allow for acclimation. After the acclimation period, the pipet tip was removed, and the subject’s movement was recorded for thirty minutes. Eight trials were performed at each tested time increment for each compound.

#### 2.3.2. Repellent Activity in Presence of Aggregation Pheromone Components 

The benzoate compounds that exhibited the longest repellency in the first series of experiments, namely MB, M2MOB, and M3MOB, were tested in subsequent experiments. The second set of experiments was identical to the first, except that a chemical attractant blend was added to the assay. Testing was performed immediately (0 h time delay) after dosing the filter paper absorbent discs and allowing for the 5 min acclimation period. The bed bug aggregation pheromone components, (*E*)-2-hexenal and (*E*)-2-octenal [42], which have been shown to exhibit attractant properties in a laboratory setting [43,44] and elicit olfactory responses [45], were used as a bed bug blended attractant in these trials. Ten minutes before adding the four absorbent filter paper discs to the arena, ten microliters of a 1:10,000 dilution in acetone blend of (*E*)-2-hexenal and (*E*)-2-octenal (1:1 *v*/*v*, 0.010%) were applied directly to the filter paper lining the Petri dish arena in the location where the repellent-treated filter paper disc was to be placed. For blank controls, ten microliters of acetone solvent were applied to the locations where the untreated filter paper discs were to be placed. Though an exact distribution could not readily be ascertained, the attractant blend (or acetone solvent only) naturally diffused through the filter paper to cover an area roughly equal to a 1 cm diameter circle (larger than the filter paper disc). After ten minutes, which was enough time to allow for acetone evaporation, the repellent absorbent disc was paced on top of the attractant and the untreated blank control discs were placed on top of the acetone treated portions of the arena. Testing then proceeded as in the previous series of experiments. The position of the repellent-treated filter paper disc was randomized across trials. Twenty trials were performed for each compound tested.

#### 2.3.3. Repellent Activity in Presence of Food Source

The final set of experiments was modified from an assay designed to investigate repellency in ticks [46]. A diagram of the ring assay used is shown in Figure 2. A 25 mm diameter circle was cut out of the center of a 47 mm diameter filter paper (Whatman qualitative filter paper, Grade 1) to create a ring of filter paper with an 11 mm border and total area of 1244 mm^2^ (12.44 cm^2^). The filter paper ring was treated with 160 μL of a repellent solution—6.25% of either MB, M2MOB, M3MOB, or DEET in acetone (10 μL of repellent total; 0.78 μL repellent per cm^2^)—or same amount of acetone as blank control. The repellent was applied in eight evenly spaced 20 μL aliquots while the filter paper ring was sitting on 8–12 pin heads to minimize contact loss. The ring was allowed to sit for 15 min to allow for acetone evaporation. 

A 150 mm diameter glass Petri dish top lined with filter paper (Whatman #1 qualitative 150 mm) was used as the test arena. In the center of the arena the cap of a 2.0 mL centrifuge tube was placed. Once the treated filter paper ring had been sitting on the pin heads for ten minutes, a *C. lectularius* that had been starved for 3–5 weeks was placed at a random location near the perimeter of the arena and covered with a pipet tip to acclimate for five minutes. After the acclimation period, the treated filter paper ring was placed around the centrifuge cap. Ten drops of warmed (39 °C) type AB negative human blood (WRNMMC blood bank; Bethesda, MD, USA) was placed in the cap as an attractive food source. The pipet tip was then removed, and the movement of the *C. lectularius* was recorded. The blood did cool to room temperature over the duration of the trial. Trials ended either when the subject first crossed over the inner border of the filter paper ring, or after forty minutes (susceptible strain) or 120 min (resistant strain).

### 2.4. Analysis

#### 2.4.1. Repellency Longevity

A two-centimeter diameter zone was defined within the EthoVision^©^ XT10 video tracking software (Noldus, Wageningen, Netherlands) (undetectable by the subject) for each of the four filter paper discs within the arena, with the filter paper disc being in the center of the zone (Figure 2A). The amount of time the subject spent in each of the four zones was recorded. The three untreated zones were averaged to provide a single value and compared to that of the treated zone. An unpaired two-tailed t-test was performed for each condition. If the average time spent near the untreated filter paper discs was significantly greater than the time spent near the treated filter paper disc, the condition was deemed to be repellent. 

#### 2.4.2. Repellent Activity in Presence of Aggregation Pheromone Components

A two-centimeter diameter zone was defined within the EthoVision^©^ XT10 video tracking software (Noldus, Wageningen, Netherlands) (undetectable by the subject) for the filter paper disc that was surrounded by an attractant and treated with a repellent. The amount of time the subject spent within this zone was recorded for all repellent compounds tested. An ordinary one-way ANOVA with a Holm-Šidák’s multiple comparisons test was performed comparing the amount of time the subject spent in the attractant zone across repellent compounds.

#### 2.4.3. Repellent Activity in Presence of Food Source 

The amount of time until the subject first crossed into the attractant zone was recorded. If a subject remained completely immobile for ten consecutive minutes (whether at the start of the trial or within the trial), it was discarded from analysis. If a subject did not enter the attractant zone by the end of a trial, then the trial time end point was used. The time end point was forty minutes for the susceptible strain, and 120 min for the resistant strain. This difference was due to behavioral differences between the strains, namely that the resistant strain was generally less active. No subject in a control (warm blood, no repellent) test condition reached the end point without entering the attractant zone. A logarithmic transformation of each data set [Y_new_ = log(Y_old_)] was conducted so the data would better conform to normality. An unpaired, two-tailed t-test was performed comparing the blank to the control. An ordinary one-way ANOVA with a Holm-Šidák’s multiple comparisons test was performed comparing how long it took subjects to enter the attractant zone in the presence of various repellents. All statistical analysis was performed using GraphPad Prism^®^ 9 (GraphPad Prism, La Jolla, CA, USA).

## 3. Results

In the repellency longevity experiments (Table 1), all compounds tested showed significant repellency (*p* < 0.05) when tested immediately (at 0 h) except for hexyl benzoate (HB) and DEET. When tested after 24 h, vinyl benzoate (VB), methyl 2-chlorobenzoate (M2CB), methyl 3-methylbenzoate (M3MB), methyl 2-methoxybenzoate (M2MOB), and DEET showed repellency. When tested at 7 days, M2MOB, M3MOB, and DEET showed repellency. Interestingly, M3MOB did not show statistically significant repellency at 24 h but did at 7 days. Further testing was performed on M2MOB, M3MOB, and DEET until the point of repellency loss was determined for the benzoate compounds. M2MOB maintained repellency through 21 days, but repellency was lost between 21–28 days. M3MOB lost repellency prior to 14 days. DEET showed repellency through the 28 days testing period. 

For tests run in the presence of aggregation pheromones, when only the aggregation mixture blend (attractant) was present around the filter paper disc, the subject spent more time near the treated filter paper disc than near the untreated filter paper discs (*p* = 0.0007), showing successful attractancy (Figure 3). When DEET, MB, M2MOB, or M3MOB were applied to the filter paper discs in addition to the attractant, the subject spent less time near the treated filter paper disc than when only the attractant was present (*p* < 0.0001 for all four compounds) (Figure 4).

Human blood was shown to be attractive to both strains when compared to either a blank or a warm water control, with the susceptible strain taking an average of 2.5 min to reach blood, 11.4 min to reach water, and 11.3 min to reach the blank (Figure 5A) [ANOVA: *n* = 20 per treatment (*n* = 60 total), F(2, 57) = 6.573, *p* = 0.0027. Holm-Šidák’s multiple comparisons: Warm blood vs. blank, *p* = 0.0025; warm blood vs. warm water, *p* = 0.0322; blank vs. warm water, *p* = 0.2959], and the resistant strain taking an average of 13.9 min to reach blood, 66.1 min to reach water, and 33.7 min to reach the blank (Figure 5B) [ANOVA: *n* = 20 per treatment (*n* = 60 total), F(2, 57) = 4.613, *p* = 0.0139. Holm-Šidák’s multiple comparisons: Warm blood vs. blank, *p* = 0.0451; warm blood vs. warm water, *p* = 0.0182; blank vs. warm water, *p* = 0.6135]. When repellent compounds are present, an ANOVA of the treatments run against the susceptible strain [*n* = 20 per treatment (*n* = 100 total), F(4, 95) = 27.07) (Figure 6A)] and against the resistant strain [*n* = 20 per treatment (*n* = 100 total), F(4, 95) = 11.77) (Figure 6B)] showed a significant difference between repellent treatments (*p* < 0.0001 for both). A Holm-Šidák’s multiple comparisons test of the susceptible strain showed repellency with all tested compounds (Figure 6A) and showed stronger repellency in M2MOB and M3MOB than in MB (Control vs. any other treatment, *p* < 0.0001; MB vs. M2MOB, *p* = 0.0294; MB vs. M3MOB, *p* = 0.0083) (Figure 6A). A Holm-Šidák’s multiple comparisons test of the resistant strain yielded similar results, with an additional significance between MB and DEET, with DEET being more repellent: Control vs. DEET, *p* < 0.0001; Control vs. MB, *p* = 0.0011; Control vs. M2MOB, *p* < 0.0001; Control vs. M3MOB, *p* < 0.0001; DEET vs. MB, *p* = 0.0442; MB vs. M2MOB, *p* = 0.0001; MB vs. M3MOB, *p* = 0.0001 (Figure 6B). 

## 4. Discussion

From these experiments, M2MOB stands out in its ability to provide repellency against *C. lectularius*, as it was the only tested benzoate compounds that maintained repellency at 7, 14 and 21 days after application (Table 1). M3MOB has the potential to exhibit long lasting repellency as well, but with seemingly less consistency, as it showed repellency at 7 days, but the repellency was not found to be significant at 24 h. With the exception of hexyl benzoate (HB), the other benzoate compounds tested—VB, EB, M2CB, M2MB, and M3MB—also showed immediate repellency. The results of the second and third experiments also confirm prior findings that DEET can act as a repellent for bed bugs [25,47] and provide DEET alternatives against bed bugs. It is likely that DEET repellency was not demonstrated in the first assay at zero hours because DEET’s low volatility reduces the distance at which it is effective [48]. In this assay, the subjects were not required to contact the repellent directly, and the zone of interest was wide enough to include an area that was likely not affected by the presence of DEET. Heatmaps indicate that repellency in the benzoate compounds extended beyond the 2 cm diameter zone analyzed (Figure 7), suggesting a large spatial attribute to repellency, but no quantitative analysis was done in this experiment. As such, an experiment testing the area of effect of these compounds is recommended for a future study. 

Because bed bug infestations are difficult and expensive to treat, preventative measures remain of the utmost importance. Moreover, a recent study suggests that pyrethroid resistant bed bug strains may show decreased aversion to DEET [49]—though it is worth noting that this finding was not confirmed in the present study, possibly because a different strain was tested or because a different experimental protocol was utilized. Because MB, M2MOB, and M3MOB were found to show similar repellency in the pyrethroid resistant strain compared to the susceptible strain, these compounds have potential to be effective in field strains. The long-lasting repellency of M2MOB and M3MOB, compounded with the findings that they are effective in the presence of an attractant, make them good candidates for preventative treatments.

Because the effect lasts at least seven days in M2MOB and M3MOB, compared to less than seven days for the other tested benzoate compounds, these two compounds would be good candidates for real-world scenarios where bed bug prevention is important, such as during travel. If luggage is being treated prior to travel, a week-long repellency would necessitate little-to-no reapplication during a typical trip, whereas other compounds would require more frequent reapplication. For short term or immediate treatment, other benzoate compounds, with the exception of HB, exhibit similar repellency to M2MOB, M3MOB, and DEET. The long-lasting repellency of M2MOB and M3MOB would also be useful in situations where the goal is to minimize bed bug spread, such as between neighboring apartments, and have potential to be used in conjunction with heat or freeze treatments, as a means of keeping the infestation within the boundaries of where the temperature treatment is being applied. If these repellents are capable of breaking up aggregation, which was not tested here but is a possibility given their success in the presence of pheromone components, these compounds could potentially be used to force bed bugs out of crevices, thus exposing them to desiccant dust or insecticide treatments used in conjunction. As generally safe compounds, particularly MB and M2MOB [50,51], there is also potential for these compounds to be applied directly to the skin to prevent bug bites; however, topical safety still must be demonstrated directly.

While the present study indicates potential for *C. lectularius* repellency in benzoate compounds, particularly M2MOB and M3MOB, further studies will help to understand their full potential. For instance, methods that closely mimic real-world situations could be performed to further test the ability of these compounds to prevent *C. lectularius* infestations during travel. Such methods could be similar to a previously described lunch bag assay [49] to mimic dirty laundry in a suitcase, or a previously described assay using stools and CLIMBUP^®^ Interceptors [26], which tests the ability of bed bugs to climb up the legs of furniture. While the findings here demonstrate a practical use for repelling *C. lectularius* with benzoate compounds, a similar study performed with varying doses of the best candidates, namely M2MOB and M3MOB, would be valuable. A future study could aim to generate a dose–response curve so that repellents can be produced with maximized efficiency. Additionally, while the present study demonstrates repellency, it does not suggest a mechanism of action. Future work should include molecular studies that aim to determine a mechanism by which benzoate compounds act as repellents.

DEET is a synthetic compound and has been used as organic solvent, which will dissolve or damage many plastics [52]. Repellent safety on painted surfaces, and various synthetic fabrics that may be damaged by DEET, should be demonstrated to ensure such treatment will not damage personal belongings. However, MB and M2MOB (also called methyl o-anisate) are natural products which have been approved by the US Food and Drug Administration [50] and the European Union [51] as food additives. Although these compounds have also been widely used as a fragrance ingredient and preservative in many personal care applications, they would still need to be registered as a repellent/pesticide with the EPA for practical application.

## 5. Conclusions

The present study investigated a botanical compound MB and its individual analogs for behavioral action on the common bed bug *C. lectularius* using video tracking software. The results demonstrated that two MB analogs, M2MOB and M3MOB, not only exhibited higher effect and extended durational repellency against *C. lectularius* than the parent compound MB, but also showed efficient repellency against a pyrethroid resistant strain of *C. lectularius.* Additionally, the present study investigated individual compounds, but it is possible that a mixture of compounds could produce a synergistic effect and increase efficiency, so mixtures should be tested. Lastly, while only M2MOB and M3MOB exhibited long-lasting (at least 7 days) repellency when applied neat, it is possible that formulations could be made that would prolong the repellent effect of the other compounds. As such, a deeper understanding of the repellent effects of all of these compounds would be valuable.

## Figures and Tables

**Figure 1 insects-13-01060-f001:**
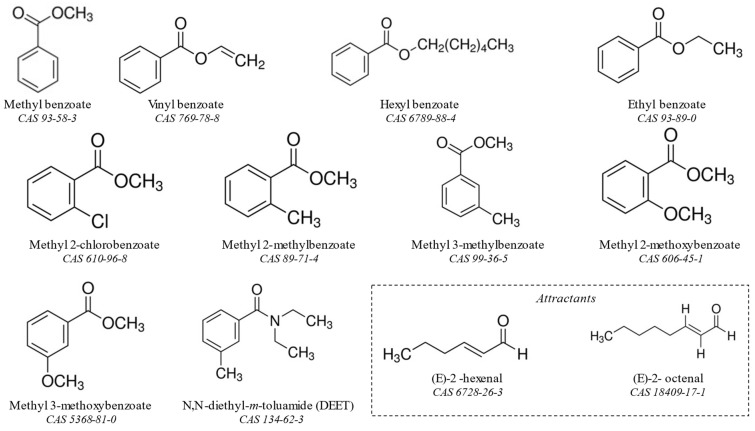
Chemical structure and CAS registry numbers of all compounds tested in this set of experiments. Methyl benzoate (MB), vinyl benzoate (VB), hexyl benzoate (HB), ethyl benzoate (EB), methyl 2-chlorobenzoate (M2CB), methyl 2-methylbenzoate (M2MB), methyl 3-methylbenzoate (M3MB), methyl 2-methoxybenzoate (M2MOB), methyl 3-methoxybenzoate (M3MOB), *N*,*N*-diethyl-*m*-toluamide (DEET), (*E*)-2-hexenal, and (*E*)-2-octenal.

**Figure 2 insects-13-01060-f002:**
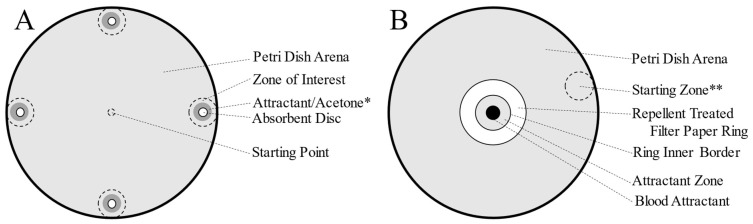
Diagrams showing the assays used in this study. (**A**) Four-disc assays (repellency longevity and repellency in presence of aggregation pheromone components). A 15 cm Petri dish was lined with filter paper, and four absorbent filter paper discs (one of which contained a repellent compound) were evenly spaced close to the perimeter of the arena. Trials began with a bed bug in the center of arena. * Attractant/acetone was present under the absorbent filter paper discs only in the second set of experiments (with aggregation pheromone components). (**B**) Repellent activity in presence of food source. A 15 cm Petri dish was lined with filter paper, and a centrifuge cap filled with human blood was placed in the center of the arena and surrounded by a ring of filter paper treated with a repellent compound. ** Starting zone was randomized, but always along the border of the arena.

**Figure 3 insects-13-01060-f003:**
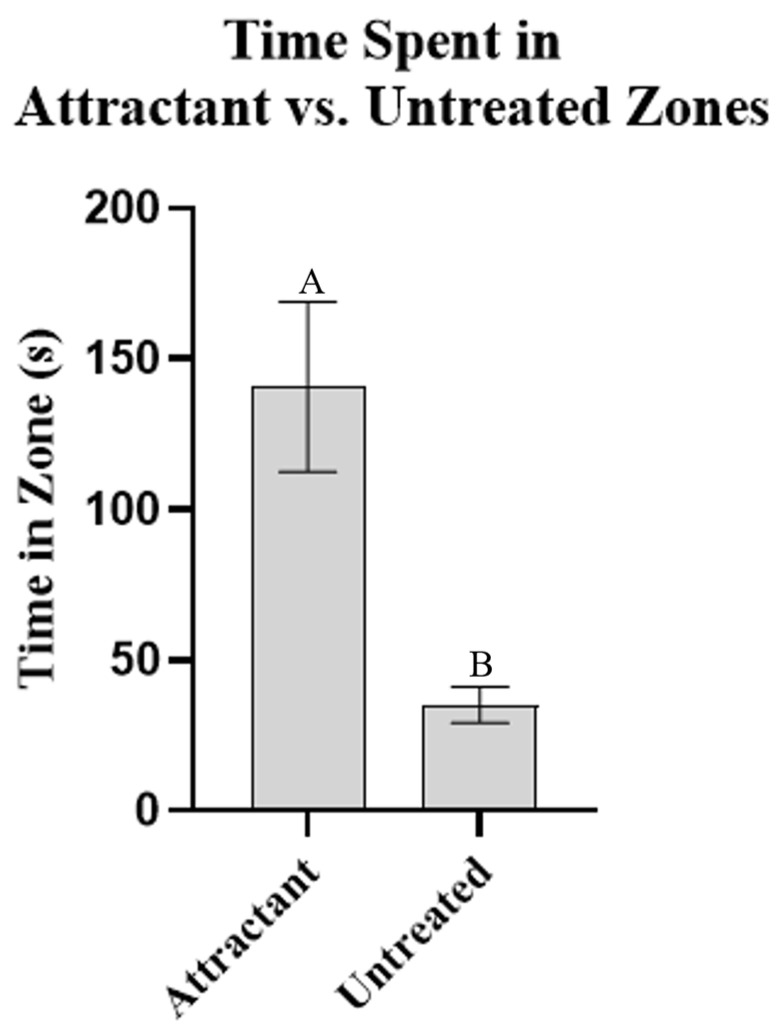
Repellent activity in presence of aggregation pheromone components, baseline results. Amount of time (mean +/− SEM) the subject spent in zone treated with attractant [equal parts by volume of (*E*)-2-hexenal and (*E*)-2-octenal, diluted 1:10,000 (0.01%) in acetone] compared to average of time spent in the three untreated zones. Different letters above them are significantly different at α = 0.05. *t*-test: *n* = 20, *p* = 0.0007.

**Figure 4 insects-13-01060-f004:**
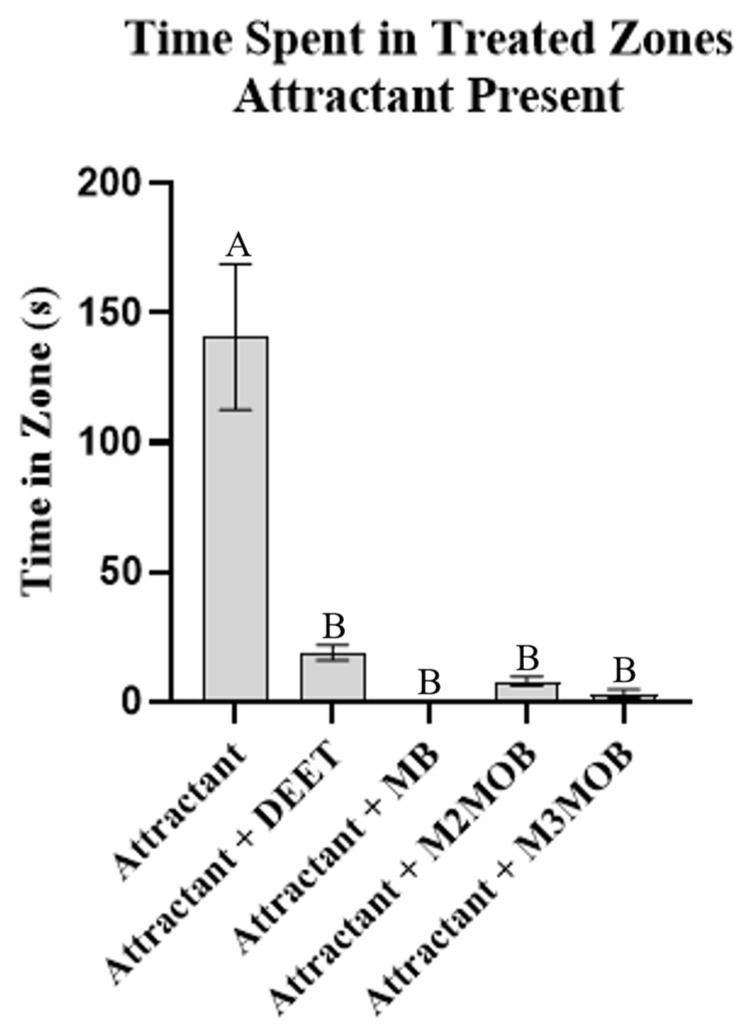
Repellent activity in presence of aggregation pheromone components. Amount of time (mean +/− SEM) the subject spent in zone treated with attractant [equal parts by volume of (*E*)-2-hexenal and (*E*)-2-octenal, diluted 1:10,000 (0.01%) in acetone] and repellent [DEET, methyl benzoate (MB), methyl 2-methoxybenzoate (M2MOB), or methyl 3-methoxybenzoate (M3MOB)]. Different letters above them are significantly different at α = 0.05. ANOVA: *n* = 20 per treatment (*n* = 100 total), F(4, 95) = 22.37, *p* < 0.0001. Holm-Šidák’s multiple comparisons: Control vs. any other treatment, *p* < 0.0001; all other comparisons not significant.

**Figure 5 insects-13-01060-f005:**
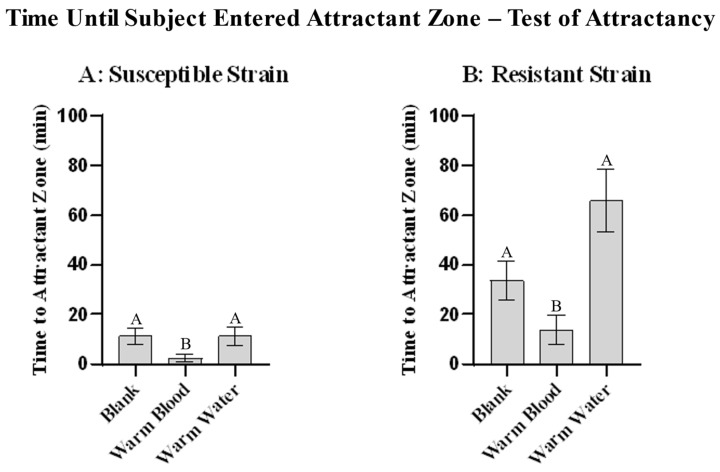
Repellent activity in presence of food source, baseline results. The amount of time (mean +/− SEM) it took for subjects in susceptible strain (**A**) and resistant strain (**B**) to enter the attractant zone, in minutes, when zone contained warmed blood, warmed water, or neither (blank). Different letters above them are significantly different at α = 0.05. A: ANOVA: *n* = 20 per treatment (*n* = 60 total), F(2, 57) = 6.573, *p* = 0.0027. Holm-Šidák’s multiple comparisons: Warm blood vs. blank, *p* = 0.0025; warm blood vs. warm water, *p* = 0.0322; blank vs. warm water, *p* = 0.2959. B: ANOVA: *n* = 20 per treatment (*n* = 60 total), F(2, 57) = 4.613, *p* = 0.0139. Holm-Šidák’s multiple comparisons: Warm blood vs. blank, *p* = 0.0451; warm blood vs. warm water, *p* = 0.0182; blank vs. warm water, *p* = 0.6135.

**Figure 6 insects-13-01060-f006:**
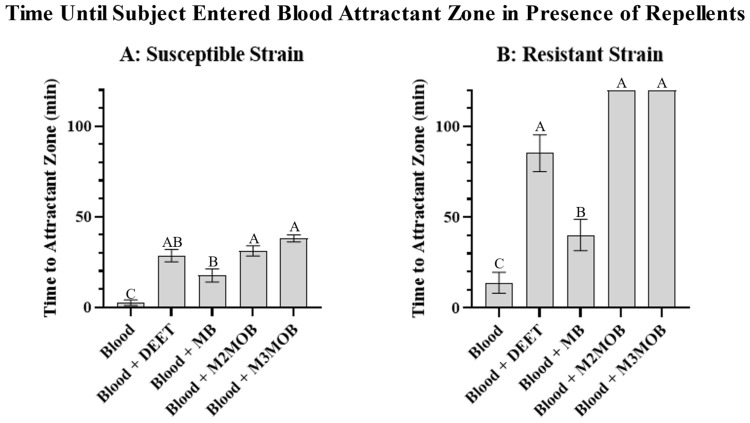
Repellent activity in presence of food source. The amount of time (mean +/− SEM) it took for subjects in susceptible strain (**A**) and resistant strain (**B**) to enter the blood attractant zone, in minutes, when zone is surrounded by a repellent compound [0.14 μL/cm^2^ of DEET, methyl benzoate (MB), methyl 2-methoxybenzoate (M2MOB), or methyl 3-methoxybenzoate (M3MOB)]. Error bars are not shown for M2MOB and M3MOB of the resistant strain because all subjects went the entire 120 min without entering the attractant zone for both treatments. Different letters above them are significantly different at α = 0.05. A: ANOVA: *n* = 20 per treatment (*n* = 100 total), F(4, 95) = 27.07, *p* < 0.0001. Holm-Šidák’s multiple comparisons: Control vs. any other treatment, *p* < 0.0001; MB vs. M2MOB, *p* = 0.0294; MB vs. M3MOB, *p* = 0.0083; all other comparisons not significant. B: ANOVA: *n* = 20 per treatment (*n* = 100 total), F(4, 95) = 11.77, *p* < 0.0001. Holm-Šidák’s multiple comparisons: Control vs. DEET, *p* < 0.0001; Control vs. MB, *p* = 0.0011; Control vs. M2MOB, *p* < 0.0001; Control vs. M3MOB, *p* < 0.0001; DEET vs. MB, *p* = 0.0442; MB vs. M2MOB, *p* = 0.0001; MB vs. M3MOB, *p* = 0.0001; all other comparisons were not significant.

**Figure 7 insects-13-01060-f007:**
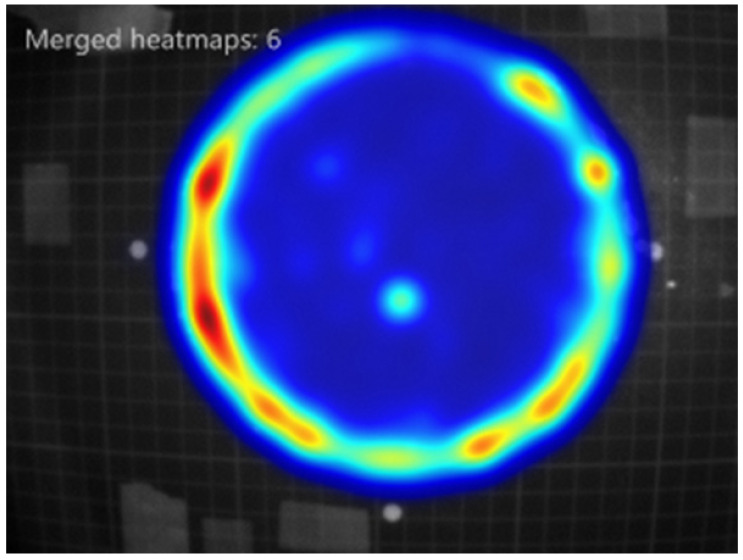
A representative heatmap showing where bed bugs within the four-disc repellency assay spent time, with cool (blue) colors indicating less time, and warm (red) colors indicating more time. The repellent treated filter paper disc was at the top position, while the three untreated filter paper discs were at the right, left, and bottom positions. This image was merged from six of the eight trials where methyl benzoate was tested after 24 h (the other two trials were performed on a second setup, and therefore not included in this file).

**Table 1 insects-13-01060-t001:** Repellency of benzoate compounds over time. If more time was spent, on average, near the untreated filter paper discs, and data were statistically significant (*p* < 0.05), the condition is labeled “R” (repellent) and text is in bold. If the time spent near the untreated discs was not significantly greater than the time near the treated disc, the condition is labeled “NSR” (no significant repellency). Repellency label and *p* value from an unpaired, two-tailed *t*-test are shown for each compound tested, ranging from immediate testing to seven days for all compounds (0 h, 24 h, 7 days). Additional tests were performed on compounds that showed repellency at 7 days, up to 28 days for M2MOB and DEET. Eight trials (*n* = 8) were run for each condition tested, but one subject in the “DEET, zero hour” condition did not enter any of the four zones and was therefore not included in the analysis (*n* = 7 for this condition). Conditions not tested are labeled “*NT*”.

	0 h	24 h	7 Days	14 Days	21 Days	28 Days
**Blank**	NSR *p* = 0.1160	NSR *p* = 0.1877	NSR *p* = 0.1572	*NT*	*NT*	*NT*
**HB**	NSR *p* = 0.3312	NSR *p* = 0.5123	NSR *p* = 0.1893	*NT*	*NT*	*NT*
**EB**	**R** ***p* = 0.0005**	NSR *p* = 0.6028	NSR *p* = 0.1247	*NT*	*NT*	*NT*
**M2MB**	**R** ***p* = 0.0007**	NSR *p* = 0.5266	NSR *p* = 0.4672	*NT*	*NT*	*NT*
**VB**	**R** ***p* < 0.0001**	**R** ***p* = 0.0174**	NSR *p* = 0.3912	*NT*	*NT*	*NT*
**M2CB**	**R** ***p* < 0.0001**	**R** ***p* < 0.0001**	NSR *p* = 0.7350	*NT*	*NT*	*NT*
**M3MB**	**R** ***p* < 0.0001**	**R** ***p* < 0.0001**	NSR *p* = 0.7953	*NT*	*NT*	*NT*
**M3MOB**	**R** ***p* = 0.0026**	NSR *p* = 0.0928	**R** ***p* < 0.0001**	NSR *p* = 0.8232	*NT*	*NT*
**M2MOB**	**R** ***p* = 0.0002**	**R** ***p* = 0.0001**	**R** ***p* = 0.0421**	**R** ***p* = 0.0022**	**R** ***p* = 0.0081**	NSR *p* = 0.7433
**DEET**	NSR *p* = 0.2705	**R** ***p* = 0.0001**	**R** ***p* = 0.0006**	**R** ***p* = 0.0397**	**R** ***p* = 0.0002**	**R** ***p* = 0.0067**

## Data Availability

The data presented in this study are available on request from the corresponding author.

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
