# Peer review of "Characterizing Repellencies of Methyl Benzoate and Its Analogs against the Common Bed Bug, Cimex lectularius"

_insects, 2022, doi:10.3390/insects13111060_

Round 1

Reviewer 1 Report (Previous Reviewer 1)

Manuscript: Characterizing Repellencies of Methyl Benzoate and its Analogs against the Common Bed Bug, Cimex lectularius

Authors: Jaime Strickland, Nicholas Larson, Mark Feldlaufer, Aijun Zhang

This manuscript is a revision of a previously submitted manuscript that was rejected. While I found the topic of this paper interesting and appropriate for this journal, the authors made only minimal revisions on the original 6 major concerns I had, rather justifying the decisions made.  While I appreciate their explanations, many of my original concerns still exist.  I think the authors would’ve been better served by re-running some of the experiments or adding additional experiments to address the concerns listed, but that was not completed for this revision.  My major concerns are restated below, with acknowledgment of the authors response.

1) Variable starvation times: The authors justify why this information may be useful, but fail to correct for the fact that variable starvation times can result in biased results.  It is also a little more clear how the assays were performed, making it even more concerning as to why they would only use bugs with extended periods of starvation for some assays.

2) Dose-Response: The authors indicate that a D-R is valuable addition, but not needed at this stage.  I disagree.  As originally stated, I believe the authors need to justify why these doses were used, which can be done based on biological relevance or label rate (for formulated products).  If this information doesn’t exist, a D-R is critical to the study and should be performed.

3) Position Randomization: The authors justify lack of randomization due to convenience in generating data summary figures.  While I understand their reasoning, this is still a major problem in experimental design.  Unfortunately, control data collected at a different time doesn’t fix this problem.

4) Were the Compounds Still Wet: Thank you for the explanation, I have no concerns regarding this anymore.

5) Use of Human Blood as an Attractant: Given the authors confidence that human blood is attractive to bed bugs, and the lack of any published literature showing blood to be attractive to bed bugs, this must be expanded in the discussion. As stated previously, the authors may not have intended this but to say that human blood is attractive to bed bugs has major implications and must be addressed if the authors believe these results to be true.  It is certainly within the scope of this paper since this paper is the first to test and report blood to be attractive to bed bugs.  That said, I urge the authors to think about the biological relevance of this and why human blood outside of the body would be attractive to an ectoparasite that feeds on hosts without skin abrasions, and if this makes sense to test.

6) Lack of DEET Repellency: This doesn’t make sense.  Why would DEET be repellent at all other times in the same arena (same size and thus same spatial dynamics) but not at 0h?  I would encourage the authors to repeat this experiment with DEET, since there is no logical explanation for why DEET would fail at 0h, but work at all other times.

Author Response

Reviewer 1

This manuscript is a revision of a previously submitted manuscript that was rejected. While I found the topic of this paper interesting and appropriate for this journal, the authors made only minimal revisions on the original 6 major concerns I had, rather justifying the decisions made.  While I appreciate their explanations, many of my original concerns still exist.  I think the authors would’ve been better served by re-running some of the experiments or adding additional experiments to address the concerns listed, but that was not completed for this revision.  My major concerns are restated below, with acknowledgment of the authors response.

1) Variable starvation times: The authors justify why this information may be useful, but fail to correct for the fact that variable starvation times can result in biased results.  It is also a little more clear how the assays were performed, making it even more concerning as to why they would only use bugs with extended periods of starvation for some assays.

Response: The authors are not entirely clear as to what kind of bias may be had within this assay using variable starvation times. The goal was to have “hungry” individuals so that they would be more inclined to approach the blood in the assay utilizing this as an attractive food source. The other assays within this study did not utilize purposefully starved bed bug individuals. Those assays utilized bedbugs that did not visually have a blood meal within them at the time of selection for testing. This was done to improve the chances that a bed bug would explore the test arena and not just sit in a single location.

2) Dose-Response: The authors indicate that a D-R is valuable addition, but not needed at this stage.  I disagree.  As originally stated, I believe the authors need to justify why these doses were used, which can be done based on biological relevance or label rate (for formulated products).  If this information doesn’t exist, a D-R is critical to the study and should be performed.

Response: A dose response is not critical for this study. This study’s intent was to provide a proof of concept that repellency could be achieved with the various compounds tested against the bed bug. A dose response would be needed to determine the lowest concentration needed to elicit repellency in bed bugs so that if these compounds could be commercialized

3) Position Randomization: The authors justify lack of randomization due to convenience in generating data summary figures.  While I understand their reasoning, this is still a major problem in experimental design.  Unfortunately, control data collected at a different time doesn’t fix this problem.

Response: Lines 110-111 explain that a blank control was utilized to verify that there was no positional bias. Track results show that individuals will explore the whole edge and most of the center of the test arena when placed into this test. The treated areas are close to the edge of the test arena and with the heat map it is clearly shown that bed bugs approach this area but do not spend much time there, whereas the rest of the arena show signs of bed bug exploration throughout.

4) Were the Compounds Still Wet: Thank you for the explanation, I have no concerns regarding this anymore.

Response: Thanks.

5) Use of Human Blood as an Attractant: Given the authors confidence that human blood is attractive to bed bugs, and the lack of any published literature showing blood to be attractive to bed bugs, this must be expanded in the discussion. As stated previously, the authors may not have intended this but to say that human blood is attractive to bed bugs has major implications and must be addressed if the authors believe these results to be true.  It is certainly within the scope of this paper since this paper is the first to test and report blood to be attractive to bed bugs.  That said, I urge the authors to think about the biological relevance of this and why human blood outside of the body would be attractive to an ectoparasite that feeds on hosts without skin abrasions, and if this makes sense to test.

Response: It is not the goal of the authors nor this paper to suggest that human blood acts as a long range attractant that bed bugs utilize for host seeking. Blood was used as a food source to elicit a response that differentiates between a control to show an increased attraction to a location within the test arena. Figure 5 shows that bed bugs will go to a small cap of blood quicker than controls, thus validating the assay used within this study. The authors are not sure what kind of implications this would have considering bed bugs are known to feed off of blood within laboratory settings without the need of other chemical cues.

6) Lack of DEET Repellency: This doesn’t make sense.  Why would DEET be repellent at all other times in the same arena (same size and thus same spatial dynamics) but not at 0h?  I would encourage the authors to repeat this experiment with DEET, since there is no logical explanation for why DEET would fail at 0h, but work at all other times.

Response: Lines 251-255 give a reasonable explanation as to why DEEt was shown to be not significantly repellent at 0 hr. It is likely that DEET repellency was not demonstrated in the first assay at zero hours because DEET’s low volatility reduces the distance at which it is effective [51]. In this assay, the subjects were not required to contact the repellent directly, and the zone of interest was wide enough to include an area that was likely not affected by the presence of DEET.”

Reviewer 2 Report (Previous Reviewer 3)

Authors have fully replied to the reviewer's criticisms. For this reason, I suggest accepting the manuscript.   

Author Response

Thanks,

Reviewer 3 Report (New Reviewer)

Characterizing Repellencies of Methyl Benzoate and its Analogs against the Common Bed Bug, Cimex lectularius written by Strickland and colleagues

Lines 33 to 35: “To remove an existing bed bug infestation, integrated pest management programs …”

It is demonstrated that desiccant dusts are harmful to human health. Please see Akhoundi et al. 2019

Akhoundi M, Bruel C, Izri A. Harmful Effects of Bed Bug-Killing Method of Diatomaceous Earth on Human Health. J Insect Sci. 2019 Sep 1;19(5):13.

Lines 37 to 39: “A study comparing two different integrated pest management strategies …”

Please indicate using which method?

Lines 55 to 57: “Most commercially available bed bug control products currently on the market ….”

Please indicate where exactly (which country)?

Lines 61 to 63: “The present study builds off this work by further investigating the repellent properties …”

What is real value of the repellents while they lead to dispersion of bed bugs to other uninfested location?

Line 90: “Experimental setup”

The schematic pattern given in figure 2 can be improved by adding supplementary information about the distance between examined specimens and repellent compounds.

Lines 100 to 102: “Ten microliters of pure test compound was applied to …”

Which concentration? Did all chemicals tested in this study have an almost equal concentration?

Lines 125 to 126: “The bed bug aggregation pheromone components, (E)-2-hexenal and (E)-2-octenal …”

The role of (E)-2-hexenal and (E)-2-octenal in bed bugs’ biology and their behavior is controversial. Some authors reported them as aggregation VOCs (volatile organic compounds) (Mendki et al. 2014, Gries et al. 2015) whereas some other considered them as alarm compounds (Levinson 1974, Haracca et al. 2010, Liedtke et al. 2011). Therefore, how the authors were sure about the role of mentioned compounds in their in vivo analysis?

Lines 131 to 133: “For blank controls, ten microliters of 131 acetone solvent …”

How the authors are sure about neutrality of “acetone solvent” as blank control?

Line 141: “Repellent activity in presence of food source”

The volume, concentration and the distance between examined bed bugs and tested compounds are important factors playing role in this olfactometric test while this information are missing in this section. Moreover, there is a serious lack of information about the physiological status of examined bed bugs, their sex, age, blood feeding (fed/unfed), and breeding (parous/nulliparous) status. Furthermore, there is no information about the environmental factors such as temperature, humidity, light, etc.

Lines 203 to 206: “When tested after 24 hours, methyl benzoate (MB), …”

Did all variants of methylbenzoate demonstrate the same amount of repellency? How the authors could measure the level of this repellency in diverse tested compounds?

 Lines 253 to 255: “In this assay, the subjects were not 253 required to contact …”

The distance between repellent compounds and exposed specimens is important but it is missing in this in vivo analysis.

Line 303: “DEET is a synthetic compound and has been used as organic solvent…”

Please add the application cases of tested repellents in the real life and field condition. Where these results can be applicable? Please discuss about the advantages and defects of this olfactometric analysis using repellents and their application in control management strategies against bed bugs.

Author Response

Reviewer 3

Lines 33 to 35: “To remove an existing bed bug infestation, integrated pest management programs …”

It is demonstrated that desiccant dusts are harmful to human health. Please see Akhoundi et al. 2019

Akhoundi M, Bruel C, Izri A. Harmful Effects of Bed Bug-Killing Method of Diatomaceous Earth on Human Health. J Insect Sci. 2019 Sep 1;19(5):13.

Response: It seems like this reviewer just wants us to add this citation, I looked it up and it is just a letter to the editor and not a study. It just compiles examples of other studies suggesting that diatomaceous earth results in silicosis and potentially lung cancer.

Lines 37 to 39: “A study comparing two different integrated pest management strategies …”

Please indicate using which method?

Response: Information has been added defining what two methods were utilized in the study.

Lines 55 to 57: “Most commercially available bed bug control products currently on the market ….”

Please indicate where exactly (which country)?

Response: This line has been changed to more accurately reflect a global representation of pyrethroid use and resistance.

Lines 61 to 63: “The present study builds off this work by further investigating the repellent properties …”

What is real value of the repellents while they lead to dispersion of bed bugs to other uninfested location?

Response: The value of having a repellent for bed bugs is described within the discussion. Please see lines 271-288.

Line 90: “Experimental setup”

The schematic pattern given in figure 2 can be improved by adding supplementary information about the distance between examined specimens and repellent compounds.

Response: If the reviewer is defining examined specimens as the individual tested bed bugs within the arena, then adding information on distance between it and the repellent compounds would not necessarily be useful to the design description. The size of the petri dish is given and starting points are clearly marked for reproducibility. Once the assay begins the bed bug does not stay stationary and thus the distance between it and the repellent compounds is constantly changing.

Lines 100 to 102: “Ten microliters of pure test compound was applied to …”

Which concentration? Did all chemicals tested in this study have an almost equal concentration?

Response: For this assay concentrations of the compounds were not used. Ten microliters of pure (neat) compound were used.

Lines 125 to 126: “The bed bug aggregation pheromone components, (E)-2-hexenal and (E)-2-octenal …”

The role of (E)-2-hexenal and (E)-2-octenal in bed bugs’ biology and their behavior is controversial. Some authors reported them as aggregation VOCs (volatile organic compounds) (Mendki et al. 2014, Gries et al. 2015) whereas some other considered them as alarm compounds (Levinson 1974, Haracca et al. 2010, Liedtke et al. 2011). Therefore, how the authors were sure about the role of mentioned compounds in their in vivo analysis?

Response: Figure 3 shows that the attractant blend works to keep an individual bed bug within the zone of detection significantly longer than compared to untreated areas.

Lines 131 to 133: “For blank controls, ten microliters of 131 acetone solvent …”

How the authors are sure about neutrality of “acetone solvent” as blank control?

Response: Acetone is consistently used within several different types of assays as blank controls. In this study acetone was used as the solvent delivery control for the various repellent solutions. The start times of these assays requiring use of acetone were delayed for 15 mins to allow for evaporation of acetone. Additionally, acetone was used to dilute both attractants and repellents and as suggested by the figures presented in this paper attractants worked as expected and the purported repellent compounds worked as expected as well. Control runs also showed that addition of acetone to discs had no effect on the traversal of the arena by the individual bugs.

Line 141: “Repellent activity in presence of food source”

The volume, concentration and the distance between examined bed bugs and tested compounds are important factors playing role in this olfactometric test while this information are missing in this section. Moreover, there is a serious lack of information about the physiological status of examined bed bugs, their sex, age, blood feeding (fed/unfed), and breeding (parous/nulliparous) status. Furthermore, there is no information about the environmental factors such as temperature, humidity, light, etc.

Response: The dimensions of the arena are given as well as positional information for the individual bed bug that would be placed within the arena for the assay. For all experiments an equal number of males and females were tested (line 78). Lines 94-99 indicate that testing occurred in a darkened room with infrared light for the recording camera to capture bed bug movement.

Lines 203 to 206: “When tested after 24 hours, methyl benzoate (MB), …”

Did all variants of methylbenzoate demonstrate the same amount of repellency? How the authors could measure the level of this repellency in diverse tested compounds?

Response: This manuscript did not measure strength of repellency. It looked at whether repellency was achieved and how long repellency lasted over a time course. In order to measure repellency strength a different set of tests would be required that would be beyond the scope of this current study.

 Lines 253 to 255: “In this assay, the subjects were not 253 required to contact …”

The distance between repellent compounds and exposed specimens is important but it is missing in this in vivo analysis.

Response: The distance between repellent compounds and exposed specimens was not determined as this was in constant flux.

Line 303: “DEET is a synthetic compound and has been used as organic solvent…”

Please add the application cases of tested repellents in the real life and field condition. Where these results can be applicable? Please discuss about the advantages and defects of this olfactometric analysis using repellents and their application in control management strategies against bed bugs.

Response: Lines 260-302 discuss the applications that repellents could provide in real world settings. Also, within this section further studies are described that could provide additional evidence for successful use of repellents against bed bugs.

Reviewer 4 Report (New Reviewer)

Dear,

The repellency of methyl benzoate and its analogs, along with the insect repellent DEET, was evaluated against the cosmopolitan insect pest Cimex lectularius. Applicable findings were achieved for this study, in which the strong repellency of methyl benzoate analogs against C. lectularius was reported. However, the following comments can enhance its quality:

Based on the journal instructions, the simple SUMMARY is absent before ABSTRACT. Also, Figures and Tables should appear after the first citation in the text.

Line 56: Add some examples of conventional pyrethroids used against bed bugs.

Lines 60-61: Add more explanations and examples for the repellent activity of methyl benzoate. It is the main part of this work.

Line 68: The number is 2. Check it throughout the text.

Line 73: For resistance was verified prior to repellent experimentation’, add more explanations: Which pyrethroid insecticide was considered? Add tested concentration and a summary of the results. How was it considered a resistance population? Add also a related reference. The reader should be able to do the same experiment.

Line 80: Remove the extra dot before chemicals. Check it throughout the text.

Lines 120-121: ‘The benzoate compounds that exhibited ... ’. The sentence is not needed. It is for results.

Line 333: ‘Cimex Lectularius’ should be italic. Check it for all scientific names.

References: The titles should be in lowercase.

Author Response

Reviewer 4

The repellency of methyl benzoate and its analogs, along with the insect repellent DEET, was evaluated against the cosmopolitan insect pest Cimex lectularius. Applicable findings were achieved for this study, in which the strong repellency of methyl benzoate analogs against C. lectularius was reported. However, the following comments can enhance its quality:

Based on the journal instructions, the simple SUMMARY is absent before ABSTRACT. Also, Figures and Tables should appear after the first citation in the text.

Response: Added: Summary: The resistance of bed bugs to many of the marketed repellents has contributed to the recent resurgence in bed bug infestations. This study tests methyl benzoate and several of its analogs for repellency against the bed bug species, Cimex lectularius. It was found that many benzoate compounds exhibit repellency against bed bugs, with methyl 2-methoxybenzoate (M2MOB) and methyl 3-methoxybenzoate (M3MOB) exhibiting the longest-lasting repellency.

Line 56: Add some examples of conventional pyrethroids used against bed bugs.

Response: Added: Pyrethroid insecticides (such as deltamethrin and lambda-cyhalothrin) have been the primary chemical control of bed bugs globally for decades, however with the development of widespread resistance their efficacy is waning [36]. (Line 62-65).

Lines 60-61: Add more explanations and examples for the repellent activity of methyl benzoate. It is the main part of this work.

Response: Changed to: “…against several insect and arachnid pests, including Drosophila suzukii, Halyomorpha halys, Plutella xylostella, Manduca sexta, Spodoptera frugiperda, Solenopsis invicta, some post-harvest pests, some stored product insect pests, and the bed bug , Cimex lectularius.”

Line 68: The number is 2. Check it throughout the text.

Response: This has been fixed.

Line 73: For resistance was verified prior to repellent experimentation’, add more explanations: Which pyrethroid insecticide was considered? Add tested concentration and a summary of the results. How was it considered a resistance population? Add also a related reference. The reader should be able to do the same experiment.

Response: Added: “For the resistant population, resistance was verified by applying 500 times the LD50 of permethrin to the bed bugs’ ventral side. Three sets of ten insects were tested with zero deaths observed.”

Line 80: Remove the extra dot before chemicals. Check it throughout the text.

Response: This has been fixed.

Lines 120-121: ‘The benzoate compounds that exhibited ... ’. The sentence is not needed. It is for results.

Response: While this sentence does summarize a result from the first part of the study, it is included here because it was the selection criteria for the following two parts of the study. In this context, we view this as being methods.

Line 333: ‘Cimex Lectularius’ should be italic. Check it for all scientific names.

Response: Done.

References: The titles should be in lowercase.

Response: Done.

Round 2

Reviewer 1 Report (Previous Reviewer 1)

This manuscript is a second revision of a previously submitted manuscript.  The authors failed to make any substantial changes as suggested and thus the comments from the previous review still need to be addressed.

Author Response

We think that the comments from the previous review have already been addressed in this version.

This manuscript is a resubmission of an earlier submission. The following is a list of the peer review reports and author responses from that submission.

Round 1

Reviewer 1 Report

Manuscript: Characterizing Repellencies of Methyl Benzoate and its Analogs against the Common Bed Bug, Cimex lectularius

Authors: Jaime Strickland, Nicholas Larson, Mark Feldlaufer, Aijun Zhang

This manuscript describes the repellent properties of Methyl Benzoate and its analogs against the common bed bug.  While I found the topic of this paper interesting and appropriate for this journal, there are several major flaws in the experimental design and method validation that cannot be corrected by a revision.  My specific concerns are listed below.

1) Why were bugs starved for 3-5 weeks (or unknown amounts for some of the assays)?  Bed bug behavior and life history can be dramatically affected by starvation (Polanco et al. 2006, Olson et al. 2009, Ahmed et al. 2021).  Specifically, the unknown time is a major concern, when bugs starved for short durations could be compared against those for long durations for different products (resulting in different results simply based on starvation time).  Unless a dose-response of repellent/attraction behavior and age can be shown, it is not appropriate to compare bugs starved for unknown amounts of time.

2) Why were certain amounts of products used (see L112, L156, 173-175, etc.)?  Are these amounts biologically relevant?  If so, please elaborate on why these doses tested.  If not, why wasn’t a dose-response performed?  A dose-response is critical if there is not biological reason for the doses tested, so that readers can determine if/how these compounds could be used (which is a central focus of the paper).  Without this, it’s hard to interpret what the results mean.  For example, at a very low dose, DEET isn’t repellent.

3) The authors failed to randomize position in experiment 1, which could seriously bias the results.  Relying on previous data to show no positional bias does not account for the assays run here.  Positional bias can often be attributed to non-constant factors (vibration from a motor running, a fan circulating air, a draft of air, etc.), which is why randomizing or rotating position is critical to ensure factors that can’t be seen are accounted for.  While the blank control provides some assurance that the bugs will visit all parts of the arenas, these are not a substitute for randomization position with the treatments (can’t account for combination effects).

4) Were the compounds tested after 0h still wet?  If so, why were they tested while still wet?  For repellency, it seems hard to imagine a scenario where you would want to test a compound that is still wet.  Without biological relevance, what value do the results of wet treatments provide?

5) Why was human blood used as an attractant?  I’m not aware of any paper showing bed bugs are attracted to human blood.  Further, what is the biological relevance of testing blood?  Bed bugs attract and orient to hosts without skin abrasions, so why use blood as an attractant as opposed to human odor?  And the current paper only shows that bed bugs find blood faster, but the time taken to find blood is extraordinarily long in field populations (>10 minutes) with no repellent.  Couldn’t attraction simply be related to the heat?  With this kind of findings and the claim that blood is an attractant, there needs to be more done including testing blood at different temperatures and water (HPLC grade) at different temperature to see what is going on here.  Although not the focus of this paper, a claim that blood is attractive is inappropriate to put forth without proper validation.

6) Why wasn’t DEET found to be repellent?  This is one of, if not the most well-studied insect repellents, which has constantly been shown to be repellent (including for bed bugs).  The lack of repellency in the first assays (0h) calls into question the validity of the first assay (DEET should be a positive control, and if a positive control fails the experiment needs to be modified or not used).

Other Comments:

L10: Should read “…animals, bed bug infestations have caused…”

L16: Consider changing this to “…have been used to control bed bugs”.  While many of these products are available for bed bug control, the vast majority are targeted towards consumers and thus not part of bed bug control programs

L23: should read “…was used to observe C. lectularius…”

L34-35: Consider expanding this.  There are lost of health effects associated with bed bugs, such as psychological, potential to transmit diseases (although not documented in people to date), microbes, new contaminants, etc.

L38-39: Who provides this recommendation?  There are lots of treatment options with good success rates other than heat and I’m not sure where this idea comes from.

L46: There are many citations form the new bed bug book.  While these are ok, it’s better to cite source material where possible.

L64-67: While not directly associated with this manuscript, it’s unclear what the focus of these products is.  In many previous papers (included those cited here), the authors lobby for the insecticidal value of these products.  However, if these compounds are highly repellent, what value do they have as insecticides (other than direct contact, where many pyrethroids remain effective as well).

Figure 4: can remove p-value from figure (stated in the captions)

Figure 5: can remove p-value from figure (stated in the captions)

Author Response

1) Why were bugs starved for 3-5 weeks (or unknown amounts for some of the assays)?  Bed bug behavior and life history can be dramatically affected by starvation (Polanco et al. 2006, Olson et al. 2009, Ahmed et al. 2021).  Specifically, the unknown time is a major concern, when bugs starved for short durations could be compared against those for long durations for different products (resulting in different results simply based on starvation time).  Unless a dose-response of repellent/attraction behavior and age can be shown, it is not appropriate to compare bugs starved for unknown amounts of time.

While it is agreed that starvation time is important, we believe the first set of tests (for which starvation time was not known) still provides useful information regarding the duration of time compounds showed repellency. In a natural environment, there is randomness in starvation time, so showing significance in a similarly random population is relevant to real world situations. This was just one of three sets of tests. Taken in conjunction with the other two assays, we believe this information is useful and strengthens the overall findings reported in this manuscript.

For all tests performed with known starvation times, the goal was to starve the subjects as long as possible, while still maintaining a population that was high enough to be practical for study. To starve subjects, they were removed from the colony and no longer provided with a food source. After five weeks, more than half of the subjects removed from the population had died, thus reducing the number of subjects available to test. A window of 3-5 weeks was found to be the ideal balance between maximizing starvation and keeping research moving forward at a practical pace.

2) Why were certain amounts of products used (see L112, L156, 173-175, etc.)?  Are these amounts biologically relevant?  If so, please elaborate on why these doses tested.  If not, why wasn’t a dose-response performed?  A dose-response is critical if there is not biological reason for the doses tested, so that readers can determine if/how these compounds could be used (which is a central focus of the paper).  Without this, it’s hard to interpret what the results mean.  For example, at a very low dose, DEET isn’t repellent.

A dose response would be a valuable addition to this research, but as this is the first study testing these compounds against C. lectularius we believe that the proof-of-concept findings presented here are still appropriate for publication. A dose response curve can be performed on these compounds in the future. Two sentences were added to the discussion indicating the value of such a future study.

3) The authors failed to randomize position in experiment 1, which could seriously bias the results.  Relying on previous data to show no positional bias does not account for the assays run here.  Positional bias can often be attributed to non-constant factors (vibration from a motor running, a fan circulating air, a draft of air, etc.), which is why randomizing or rotating position is critical to ensure factors that can’t be seen are accounted for.  While the blank control provides some assurance that the bugs will visit all parts of the arenas, these are not a substitute for randomization position with the treatments (can’t account for combination effects).

Positions were not randomized in experiment 1 because it would make a useful merge of heatmaps impossible to generate within the software. At the onset, we believed we would utilize the heatmaps more than we did, so we decided it would be valuable to keep the zones in the same position for each trial and refer to our previous study which showed no bias. We believe with the inclusion of a blank control we have strong enough data to demonstrate repellency, despite not randomizing the position in the first set of experiments.

4) Were the compounds tested after 0h still wet?  If so, why were they tested while still wet?  For repellency, it seems hard to imagine a scenario where you would want to test a compound that is still wet.  Without biological relevance, what value do the results of wet treatments provide?

In experiment 1, the compounds were applied “neat” (not dissolved in a solvent) [line 116]. As such, the “wetness” of the discs is merely a measure of how much of the compound remains. A dry disc would indicate that the compound is no longer present, and would, therefore, not be appropriate for testing.

In experiment 2, the attractant compounds were dissolved in acetone, but the acetone was allowed to evaporate before tests were run. No solvent remained at the time of testing. A sentence was added stating this more clearly [line 149].

In experiment 3, the repellents were dissolved in acetone, but the acetone was allowed to evaporate before tests were run. No solvent remained at the time of testing. [lines 163]

5) Why was human blood used as an attractant?  I’m not aware of any paper showing bed bugs are attracted to human blood.  Further, what is the biological relevance of testing blood?  Bed bugs attract and orient to hosts without skin abrasions, so why use blood as an attractant as opposed to human odor?  And the current paper only shows that bed bugs find blood faster, but the time taken to find blood is extraordinarily long in field populations (>10 minutes) with no repellent.  Couldn’t attraction simply be related to the heat?  With this kind of findings and the claim that blood is an attractant, there needs to be more done including testing blood at different temperatures and water (HPLC grade) at different temperature to see what is going on here.  Although not the focus of this paper, a claim that blood is attractive is inappropriate to put forth without proper validation.

Human blood was hypothesized to be attractive, and so was tested. Our findings indicate that bed bugs do, in fact, navigate to blood faster than to a water control. We agree that the warmth of blood could have been a factor, so we tested warm water to confirm our findings. We found that C. lectularius orients to warm blood faster than to warm water. There was no difference between warm water and room temperature water. The focus of this paper was to test the repellents in the presence of attractants, and since we showed warm blood to be attractive, this is what we tested against our repellents. We do not feel as though we have overstated the significance of the findings, but rather confirmed that blood was sufficient to test against our repellents. A full discussion of the relevance of being attracted to human is beyond the scope of the present research.

To the point of taking a long time to reach the attractant: The resistant strain tested was minimally active in all conditions, and this seems to be a trait of the specific strain itself. While the low activity level may not translate to most field populations, the important fact here is that the strain is resistant to pyrethroids but not resistant to the benzoate compounds tested here. Therefore, we believe we have demonstrated that these compounds can be effective on pyrethroid-resistant strains.

6) Why wasn’t DEET found to be repellent?  This is one of, if not the most well-studied insect repellents, which has constantly been shown to be repellent (including for bed bugs).  The lack of repellency in the first assays (0h) calls into question the validity of the first assay (DEET should be a positive control, and if a positive control fails the experiment needs to be modified or not used).

DEET was not found to be repellent in the first assay at zero hours because the assay included a zone of interest outside of the effect zone of DEET. DEET does not have a strong enough spatial component to exhibit repellency at the distance and dose tested in this experiment. This explanation was added to the discussion [lines 258-261].

Other Comments:

L10: Should read “…animals, bed bug infestations have caused…”

In addressing one of reviewer 2’s concerns, this line has been removed entirely.

L16: Consider changing this to “…have been used to control bed bugs”.  While many of these products are available for bed bug control, the vast majority are targeted towards consumers and thus not part of bed bug control programs

In addressing one of reviewer 2’s concerns, this line has been removed entirely.

L23: should read “…was used to observe C. lectularius…”

Word “the” has been removed.

L34-35: Consider expanding this.  There are lost of health effects associated with bed bugs, such as psychological, potential to transmit diseases (although not documented in people to date), microbes, new contaminants, etc.

Expanded to include psychological distress.

L38-39: Who provides this recommendation?  There are lots of treatment options with good success rates other than heat and I’m not sure where this idea comes from.

This has been reworded to incorporate other methods (with new references included).

L46: There are many citations form the new bed bug book.  While these are ok, it’s better to cite source material where possible.

We believe these citations are okay to use here, primarily because they discuss background information, rather than studies that directly lead to the content being investigated in this paper.

L64-67: While not directly associated with this manuscript, it’s unclear what the focus of these products is.  In many previous papers (included those cited here), the authors lobby for the insecticidal value of these products.  However, if these compounds are highly repellent, what value do they have as insecticides (other than direct contact, where many pyrethroids remain effective as well).

The value of repellent compounds is covered in the discussion [lines 276-291]

Figure 4: can remove p-value from figure (stated in the captions)

p-value removed. Replaced with letters indicating significance.

Figure 5: can remove p-value from figure (stated in the captions)

p-value removed. Replaced with letters indicating significance.

Reviewer 2 Report

Comments are attached as a word-file

Author Response

1) The technical description of the bioassays and the prerequisites for testing is scattered around in the manuscript (L23, L71-73, L110-112, L120-123, L125-130, L152-155, L160-163, L169-170, L224-225, L256-259, L263-267, Figure 3). The results presented in Figure 4 and Figure 6 are in my opinion the baseline of the bioassay and can therefore be given briefly in the bioassay-description without the figures (in other words: 1. attraction was confirmed without repellents present and 2. there was no positional bias). I suggest that all these elements are moved to section 2.4 (behavioral registration and analyses… ???).

Line 23 kept, because this is in the abstract and just briefly mentions the software used in this study.

Lines 71-73 removed: This belongs in the methods and is already discussed there.

Lines 125-130 combined with lines 110-112 at the start of the paragraph, but lines 120-123 were kept in the same location because it references treatment of the disc, which had not been discussed until this point in the paragraph.

Lines 152-155 and 160-163 are within the same paragraph, and we believe this reads in a logical way. Lines 152-155 were reworded, but they introduce the compounds used as attractants, and are followed by how the attractants were applied. Once the application is described, lines 160-163 explain the spatial distribution of the attractants. Any attempt to combine these lines would compromise the logical flow of this paragraph.

Lines 169-170, and the rest of that paragraph, describe a separate assay, so should not be mentioned in the other sections.

In summary, some changes were made in the methods sections to combine technical aspects within sections and to avoid repeating information across sections; however, because three different tests were utilized, we believe it is important for clarity to include the specific details of the individual tests within their respective sections, rather than attempting to combine all details into a single section.

Lines 224-225 and figure 3 have been moved to the discussion (and figure 3 is now figure 7), because while the heatmap is not hard results, it does imply a spatial component to these repellents. It was added to the discussion alongside a recommendation for spatial studies in the future.

Prerequisites for testing (lines 256-259, 263-267) are included in the results because analysis was needed to confirm that prerequisites were met. We believe these data and figures are important to convince readers that our tests are valid, particularly since these exact tests are not in the literature; however, if they are deemed unnecessary these figures can be removed and the prerequisites can be confirmed more briefly.

3) The names of the experiments/bioassay are not descriptive and intuitive. I suggest changing this to something in the direction of 2.3.1 - Repellency screening with neat compounds, 2.3.2 – Aggregation disruption and 2.3.3 - Attraction blocking… ???

“Four discs assay time varied compound screen” changed to “Repellency Longevity”

“Four-disc assay with chemical attractants” changed to “Repellent activity in presence of aggregation pheromone components”

“Ring assay with human blood attractant” changed to “Repellent activity in presence of food source”

Specific comments/problems:

Abstract

L8-18: Half the abstract is written as an introduction. This should be narrowed down. Some parts can be included in the actual introduction.

Much of the abstract has been removed, in accordance with this suggestion.

L19-L28: Results can be described more precisely

Results in abstract have been reworded for clarity. [lines 40-44]

Introduction

L32-34: This sentence does not contribute much

Removed the part of the sentence about caves and villages.

L38-40: recommended removal of bedbugs also includes other methods like desiccant dust, insect pathogenic fungi or steam treatment in an IPM strategy. This part can surely be expanded.

This has been expanded to be more inclusive.

L67-71: the numbered point 3) can be removed by writing “pesticide resistant and susceptible bed bugs” in the previous sentence

Advice taken. The inclusion of “pesticide susceptible and resistant strains” was added to point 2, and point 3 was removed. [Lines 81-83]

M&M

L79-83: The “name and address” of the lab can be removed.

We believe it is useful to include the source of the blood, similar to the inclusion of the source of chemical compounds. However, if the editor wishes to omit this information that can be done.

L83-88: This is a long explanation for a minor problem. I suggest writing specific starvation time or unknown in 2.3.1, 2.3.2 and 2.3.3

This was reworded to simply include that there was a starvation time for assays that included a food (blood) component, and has been moved to the appropriate section of the methods. We do not believe the starvation time is necessary for the other assays.

L88-91: this can also be moved to their respective experiments

Shortened this to simply state that an equal number of males and females were utilized in all of the experiments. Exact number of subjects tested are already cited elsewhere in the paper.

L92-107: These sentences are hard to read… ? Suppliers can be named briefly in the figure legend and the CAS-number and purity can simply be added to the figure. The entire section (and much of 1

the figure text) can be removed if the names of the compounds are given as suggested below. Give reference to figure 1 in L112

Methyl benzoate - MB

(CAS: 93-58-3, 99≥%)

CAS registry numbers removed from text and added to figure legend. Purities simplified by changing individual purities to “Purity of all chemicals was ≥95%.”

Reference to figure 1 added.

L109: Please use a more intuitive name

Changed to “Duration of repellency.”

L11-120: The o’clock can be removed. With reference to figure 2, top, left, right and bottom is sufficient.

Advice taken.

L122-123: If a “no-bias-test” is mentioned. Results should also be given (perhaps in 2.4)

There was not a separate test for positional bias beyond the previous study, which is cited. This line is referring to the fact that if the results of this experiment could be attributed to position, then there would be no difference between the repellents and the blank.

L142: The time aspect has already been mentioned. Perhaps parts of L133-114 and L142-145 can be combined.

The bulk of lines 142-145 were moved up and combined with lines 113-114.

L147: Please use a more intuitive name

Changed to “Aggregation disruption.”

L149: Why MB? Based on Table 1, I can not see the difference between MB, M2CB and M3MB

While these results suggest similar efficacy between MB and several other compounds tested here, we elected to study MB further because, of the benzoate compounds tested here, it is the compound with the most prior research. We view it as valuable to expand upon our knowledge of this compound because it could be useful for future studies to have a benzoate compound for which there is a breadth of knowledge.

L157: Why 0.01%? I guess to avoid the signal to act as an alarm pheromone? Reference/explanation needed?

Preliminary tests were performed, not published here, and this was the most attractive concentration.

L168: Please use a more intuitive name

Changed to “Attraction blocking with repellent ring.”

L170: Reference in wrong format

This has been fixed.

L174: Why 1.125%? Reference/explanation needed?

1.125% was a miscalculation that has now been caught. The true dilution is 6.25%. This dilution was selected so the total compound applied in the ring assay was the same as what was applied in the disc assay. Disc assay: 10ul neat. Ring assay: 160ul at 6.25% = 10ul of test compound.

L198-201: This is not at all clear for me? I guess the letters are used in some kind of way in table 1? But there I find other letters as well? No time values are given. It appears to be some kind of subjective evaluation? Was the total arena divided in four quarters? How is time near the sone defined? Numbers needs to be given somewhere in the results if these aspects are used to determine repellent ability.

The text of these lines and the table description were changed for clarity. All conditions are either called Repellent (R) or Not Significantly Repellent (NSR). Reference to figure 2A added.

Results

All three paragraphs start with a sentence pointing at figures or tables. These three sentences can be deleted and reference to figures/tables incorporated in the text.

Advice taken. First sentences removed in each paragraph, figures incorporated into the text.

L226-236: It is difficult to follow the logic of the choice for further testing (see previous comment (L149)). Maybe this is a result of the letters supplying some additional information (R, A, NR, NRA) but I am unable to interpret this. What does NRA mean (not repellent attractant ??? ?) Again, how was this evaluated? Please provide some numbers and a better explanation of how this was done…

This has been resolved by resolving other concerns. Analysis text and table description have been changed to be more clear. We do not believe adding the raw numbers is necessary, and will simply add confusion to the table because each box would then have three values (treated zone, untreated zones, p-value).

L237-242: This is not results.

Heatmap figure has been moved to discussion.

L243-247: This figure is not necessary.

Because other reviewer was concerned about the validity of our test, we feel it is necessary to include this figure, as it validates our procedure.

L24-254: on the x-axis: instead of control use attractant (or aggregation signal) and instead og DEET use attractant + DEET… and so on… This will make the figure easier to understand.

Advice taken.

L271 and L272: Spatial repellency… This bioassay reports time. I guess it could be argued that there is similarity between the two parameters in this bioassay, but time until entering sone does not say much about positions. I believe a distinction between time and position is valuable (and a potential topic for the discussion?)

After giving this more thought, we believe it is best to not make such a strong claim of “spatial” repellency in this manuscript. While the nature of the first two tests were such that the attractant could be reached without coming into direct contact with the treated discs, and thus a spatial component to the repellency can be implied, we did not test spatial aspects directly and feel that claiming a strong spatial repellency would be overstating our findings. Instead, we have moved the heatmap figure to the discussion, and state that spatial repellency seems likely and would be an interesting topic of future research.

L273-74: and L275-277: It is unnecessary to report all values of the significant post-hoc tests. The letters in figure 7 is sufficient for discrimination (simply use letters and discriminated by p<0.05 in the figure legend)

Advice was considered, but we feel many readers like to see the significance level at a glance.

L275: similar… the results are the same (C, AB, B, A, A). There is only difference in time to enter (also a potential topic for the discussion?).

See comment above.

L278-282: This figure is not necessary.

Because other reviewer was concerned about the validity of our test, we feel it is necessary to include this figure, as it validates our procedure.

L283: Replace control with attractant, and DEET with attractant + DEET… and so on… This will make

the figure easier to understand

Advice taken.

L287289: explanation for SE/SEM is not necessary. You can simply write [the average time (±SE) it

took… ]

“Error bars represent standard error” removed from all figure legends. Legends now include “mean +/- SEM” instead.

Suggestions to table 1: The letters needs to be better explained if they are to be kept in the table. I guess the critical test is at 7 days. This should also be stated more clearly in the text. It is quite natural that neat compounds recently deposited can affect behaviour in surprising ways. 0-24 hours is therefore of less value compared to the 7-day measurement. (this should also be pointed out in the discussion

– your study indicate that repellency needs to be evaluated a few days after application for reliable results). As the table is presented now, I assume that the order of compounds is based on chemical structures? Perhaps it is better to organize them according to measured repellency. Suggested order of compounds: Blank, HB, EB, VB, M2MB, M2CB, M3MB, M3MOB, M2MOB and DEET. In other words, from nothing to maximum (L309-311 can be removed)

Rows of the table have been reordered to: Blank, HB, EB, M2MB, VB, M2CB, M3MB, M3MOB, M2MOB, DEET. Other concerns addressed in previous comments.

Discussion

The discussion is a bit incomplete and needs some rewriting.

The discussion has been expanded. There is now a greater distinction made between M2MOB and M3MOB, and more of a discussion of practical implications of the findings.

It should be pointed out much more uncertainty with respect to M3MOB. It does not appear as reliable as M2MOB. It is also described as long lasting even if significant repellency only was found at 0 and 7 days. The way the discussion is written the two appear more or less equal even though the two bioassays confirming function was performed just after application where almost all compounds showed repellency. These uncertainties should be fully discussed.

The wording has been changed to emphasize M2MOB more than M3MOB and to highlight the inconsistency of M3MOB (namely, its insignificant repellency at 24 hours).

I also believe that the repellency (2.3.1), reduced aggregation (2.3.2) and increased time for locating the odor source (2.3.3) can be more fully discussed from an applied perspective. There is brief mention of luggage protection, dispersal prevention and some dim sentences regarding trapping. What about push-pull options? Reduced aggregation for increased exposure to pesticides such as desiccant dusts? Prevention of bites? Braking up hidden aggregations? As a mobility tool for forcing bed bugs out of cracks and crevises? I guess this part can be expanded quite a bit.

The discussion has been expanded to include several of these suggestions.

The similarity in response between Harland and a more natural strain should also be discussed. It is a good thing that the response is exactly the same only with a different magnitude. This indicates that the compounds are robust repellents with field potential.

Added sentence: “Because MB, M2MOB, and M3MOB were found to show similar repellency in the pyrethroid resistant strain compared to the susceptible strain, these compounds have potential to be effective in field strains.”

Potential mechanisms of the repellency should be discussed. I guess the chemical structures on these compounds are quite similar to DEET?

The mechanism is unknown. It has been suggested as a future area of research.

The brief note on mixtures and potential synergies is also interesting to follow a bit further. Natural insect repellents have been scientifically described frequently the last years. Do these products overlap with your compounds and to what extent?

We have cited the relevant papers that studied methyl benzoate as a repellent, and to our knowledge the other benzoate compounds studied here have not been thoroughly studied.

Are there any suggestions for mixing the compounds tested in this study? Further study recommendations?

There are no specific suggestions beyond mixing benzoate compounds. It’s possible mixing the stronger compounds (M2MOB, M3MOB) would be beneficial, but equally possible that mixing one of the stronger compounds (e.g. M2MOB) with one of the weaker compounds (e.g. MB, EB,…) would have a synergistic effect. Any specific suggestion would merely be speculation, and would therefore not be appropriate to include.

The conclusion should be brief and state that M2MOB is interesting and should be studied further.

We believe that the conclusion, as written, hits the major points briefly, and to remove these points would not accurately reflect the total conclusions of the paper.

Reviewer 3 Report

The common bed bug, Cimex lectularius L., is a serious is a hold nuisance. In this study, authors found several methyl benzoate analogs and they exhibited strong spatial repellency against C. lectularius. It is important in this field because authors find new natural based repellents. Most commercially available bed bug control products currently on the market develop resistance within bed bugs their efficacy is limited. This is the first study on benzoate compounds and their biological activity against C. lectularius. Four-disc assay with chemical attractants and experiment designs are new and authors statistically reported the results and conclusions. Authors used proper references. Tables and figures are clear.

Author Response

No concerns